# Social Determinants of Health and Impact on Screening, Prevalence, and Management of Diabetic Retinopathy in Adults: A Narrative Review

**DOI:** 10.3390/jcm11237120

**Published:** 2022-11-30

**Authors:** Dhruva Patel, Ajaykarthik Ananthakrishnan, Tyger Lin, Roomasa Channa, T. Y. Alvin Liu, Risa M. Wolf

**Affiliations:** 1Division of Endocrinology, Department of Pediatrics, Johns Hopkins University School of Medicine, Baltimore, MD 21205, USA; 2Department of Ophthalmology and Visual Sciences, University of Wisconsin, Madison, WI 53705, USA; 3Department of Ophthalmology, Wilmer Eye Institute, Johns Hopkins University School of Medicine, Baltimore, MD 21287, USA

**Keywords:** disparities, social determinants of health, diabetic retinopathy

## Abstract

Diabetic retinal disease (DRD) is the leading cause of blindness among working-aged individuals with diabetes. In the United States, underserved and minority populations are disproportionately affected by diabetic retinopathy and other diabetes-related health outcomes. In this narrative review, we describe racial disparities in the prevalence and screening of diabetic retinopathy, as well as the wide-range of disparities associated with social determinants of health (SDOH), which include socioeconomic status, geography, health-care access, and education.

## 1. Introduction

Diabetic retinal disease (DRD), including diabetic retinopathy (DR) and diabetic macular edema (DME), is a microvascular complication of diabetes mellitus and the leading cause of blindness among working-aged adults in the United States [1,2,3,4]. Over 40,000 new cases of DR are diagnosed each year in the US; 40.3% of American adults with diabetes over the age of 40 years have DR, while 8.2% of US adults with diabetes have advanced vision-threatening DR [5]. In the US, underserved and minority populations have worse diabetes-related health outcomes, which is likely due to a variety of factors including social determinants of health that encompass socioeconomic status, education, geography and living conditions, and access to health care and screenings. Racial inequities in the prevalence and screening of diabetic retinopathy are well described; therefore, in this narrative review, we focus on the wide-range of disparities associated with social determinants of health (SDOH) [6,7], including socioeconomic status, geography, health-care access, and education.

## 2. Methods

A literature search was conducted through Pubmed, MEDLINE, EMBASE, and Scopus from February 2022 to September 2022. The following keywords and terms were used to identify articles for review and inclusion: screening, diabetic retinopathy, diabetic retinal disease, diabetic macular edema, prevalence, management, income, education, clinical trials, disparities, social determinants of health, and access. Since the focus of the narrative review was on social determinants of health in the U.S., all included articles were in English.

While many reports have shown that minority populations have a greater prevalence of DR [8,9,10], few have explored how social determinants of health (SDOH), encompassing geographic, educational, and socioeconomic and financial barriers, influence DR prevalence and outcomes [6]. We present a summary of the reported SDOHs related to diabetic retinopathy. Due to limited published research on the impact of food, environment, and social context on diabetic retinopathy, these are not discussed in this review.

## 3. Prevalence of Diabetic Retinal Disease

Inequities in the prevalence of diabetic retinopathy exist in the US, where ethnic minority population groups have a disproportionally higher prevalence of diabetes and DR. A multi-cohort meta-analysis of National Health and Nutrition Examination Survey (NHANES) data showed that non-Hispanic blacks and Mexican American individuals had an overall higher prevalence of DR, with the prevalence of vision-threatening DR being 190% higher in non-Hispanic Black individuals and 130% higher in Mexican Americans with diabetes than in non-Hispanic white individuals [2]. Further, the third NHANES found that the prevalence of DR was 46% greater in non-Hispanic blacks with type 2 diabetes than in non-Hispanic whites in a national representative sample of adults aged greater than 40 years [11]. The LALES (Los Angeles Latino Eye Study) study of 6357 Latino participants found that 22.9% had diagnosed or undiagnosed diabetes, and of these Latino patients with diabetes, 46.9% had DR. Further, the presence of severe vision-threatening DR was found to be higher in Latinos than in non-Hispanic whites [9]. These results were replicated in a Hispanic population of 4774 individuals in New Mexico in the Proyecto Vison Evaluation Research (VER) study; the diabetes prevalence rate of 22% was almost twice that of non-Hispanic whites. Of the Hispanic patients with diabetes, the prevalence of DR was 48%, 32% of whom had moderate-to-severe non-proliferative and proliferative DR [10]. The New Jersey 725, a study of 725 African American patients with type 1 diabetes in New Jersey, found a prevalence of 63.9% for DR and 19% for proliferative DR in this cohort [12]. The Salisbury Eye Evaluation (SEE) study, a longitudinal study of 2520 adults aged 64 to 84 over a 10-year period in Maryland, found that African Americans had a 4-fold greater risk of visual impairment due to DR [8,13]. While the prevalence of diabetes is known to be higher in Alaskan Indian (AI) and Alaskan Native (AN) populations, studies have also reported significantly higher rates of non-proliferative DR in this population, which is comparable with the LALES study; for example, 37.8%, 40%, and 45.3% in Pima Indians, Navajo Indians, and the Sioux Indian tribe, respectively, have non-proliferative DR [14].

While there are significant ethnic disparities in DR prevalence, DR is a multi-faceted disease and prevalence is also influenced by several other SDOH, such as socioeconomic status, physical environment, and health care access.

### 3.1. Socioeconomic Status (Income and Education)

The prevalence of DR is higher in low-income populations with diabetes, which is due, in part, to multiple factors, including limited access to screening, low rates of follow-up visits, and a lack of insurance coverage [15,16,17,18]. A study in Chicago demonstrated that there are high rates of diabetes, and thus suspected DR, in low-income, minority communities with individuals who do not have any insurance coverage or have public insurance coverage [15].

Despite universal health coverage, studies in the United Kingdom (UK) have also shown that socioeconomic deprivation is associated with decreased participation in eye screening and diminished routine follow-up for screening [16]. This can contribute to delayed diagnoses and worse outcomes from eye disease such as macular edema, glaucoma, and retinopathy [17]. For example, one study reported a higher prevalence, and later presentation, of sight-threatening proliferative DR among socioeconomically deprived patients [18]. Another study of 150 individuals with type 1 diabetes in Spain found that patients with onset of diabetes >18 years of age and in areas of lower socioeconomic status were 3-fold more likely to develop DR compared to patients with onset of diabetes <18 years and not in areas of lower socioeconomic status [17]. Such results suggest the need to further study the role of socioeconomic status on the prevalence of DR in populations in the USA, as implementing programs to increase screening in areas of low socioeconomic status may improve DR outcomes.

Additionally, patients with lower educational attainment have been shown to have diminished adherence to eye care visits [2]. Analysis of NHANES data from 1998 to 2008 found that patients with less than a high school level of education had a significantly higher prevalence of DR compared to patients with greater than a high school level of education [19]. The New Jersey 725 also found that the frequency of visual impairment due to DR was inversely associated with educational level and employment status, suggesting that educational background may contribute to disparities in DR prevalence [12].

### 3.2. Geography (Neighborhood and Physical Environment)

Studies have also analyzed the effect of neighborhood environment on the prevalence of diabetes and DR. In a retrospective study of 7288 patients in California with type 2 diabetes, individuals residing in lower socioeconomic neighborhoods had higher HbA1c levels, which is a risk factor for DR [3,20]. Diex Roux et al. also studied the effect of neighborhood environments on diabetes in patients in the Coronary Artery Risk Development in Young Adults (CARDIA) study and found that a neighborhood socioeconomic score, which encompassed factors such as median household income, educational attainment, and occupational status, was inversely related to insulin resistance syndrome, which can result in diabetes [21]. Studies have also shown that socioeconomically deprived neighborhoods are associated with decreased uptake of DR screening [22,23]. These results suggest that residential environments play a significant role in the prevalence and treatment of diabetes and diabetes-related complications, perhaps by influencing diet and physical activity [24].

### 3.3. Healthcare Access and Affordability

Access to health care is influenced by several factors, such as geographical distance from health care facilities and insurance status; these factors are shown to influence DR prevalence [15,16,17,25,26]. In a study analyzing data from the 2006 Behavioral Risk Factor Surveillance System (BRFSS), it was found that the prevalence of diabetes and DR was greater in rural residents compared to urban residents. Further analysis of patients living in rural environments showed that although uninsured persons had lower odds of obtaining eye exam services within the past year (indicating they did not receive an eye exam in the past year), they did not have greater odds for retinopathy. However, low-income status only (under $25,000) was associated with greater odds of having DR [27]. Greater DR prevalence in rural areas may be explained by decreased access to screening services and access to care. In a study analyzing 200 records of patients with diabetes without known diabetic retinopathy, the overall eye screening compliance rate was 31% [28]. The analysis showed that patients within 8 miles of a screening center were significantly more likely to attend their screening appointments compared to those greater than 8 miles. Additionally, adherence was positively correlated with access to public transportation [28].

Lack of insurance coverage can also impede access to health care, which can result in greater DR prevalence. In a retrospective analysis of medical records of patients with diabetes in low-income areas of Chicago with higher-than-expected rates of DR, it was found that 37% of patients with diabetes were uninsured [15]. Of the patients with diabetes that were insured in these areas of Chicago, 32% were covered by Medicare and 10% were covered by Medicaid [15]. Despite having insurance, lack of coverage for specific services and reimbursement for DR screening may be a barrier to getting screened for DR. In another study analyzing follow-up appointment compliance of patients presenting to an ophthalmology emergency room with proliferative DR, public insurance was correlated with decreased follow-up [29]. This result was replicated in a separate study of patients with DM residing in rural Vermont, where poor follow-up was associated with lower socioeconomic status, as defined by Medicaid enrollment [30]. These results suggest that insurance coverage is associated with reduced follow-up for proliferative DR, and that there needs to be a greater focus for Medicare and Medicaid patients to have more access to DR screening and treatment services.

### 3.4. SDOH, Ethnicity, and Risk Factors for Diabetic Retinopathy

The strongest risk factors for DR are elevated HbA1c levels, longer duration of diabetes, elevated albumin excretion rate (AER), and higher mean diastolic blood pressure [25]. Apart from duration of diabetes, these risk factors are modifiable and exhibit inequities related to SDOH, and thus can be targeted in preventive interventions to decrease diabetes-related microvascular complications in vulnerable populations [26].

#### 3.4.1. Elevated HbA1c

The Diabetes Control and Complications Trial (DCCT) study established that in patients with insulin dependent diabetes mellitus, intensive insulin-therapy regimens with the aim of maintaining glycemic control as close to the normal HbA1c range as safely possible significantly decreases the risk of developing DR by 76%; this risk reduction also increased with time [26]. Additionally, intensive insulin therapy was also shown to slow the progression of DR by 54% and reduce progression to proliferative or severe non-proliferative retinopathy by 47% in participants that entered the study with mild retinopathy. The Action to Control Cardiovascular Risk in Diabetes (ACCORD) Eye study found that intensive treatment of hyperglycemia to bring target HbA1c levels <6% resulted in a reduced rate of progression of DR in patients with type 2 diabetes [31].

Several studies have analyzed the association between glycemic control and HbA1c levels on diabetic retinopathy in minority and underserved groups. In the US, African Americans with diabetes have higher HbA1c levels compared to non-Hispanic whites with diabetes [32]. Furthermore, the NHANES III study found HbA1c values and DR prevalence to be significantly greater in non-Hispanic blacks and Hispanics compared to non-Hispanic whites [11]. The LALES study, which recruited subjects from a Latino population, found that in participants with diabetes, every 1% increase in HbA1c (up to a value of 11%) was associated with a 22% greater risk for developing DR [8,9]. Mexican Americans with diabetes have also been shown to have more severe hyperglycemia than non-Hispanic whites; in the Proyecto VER study, the severity of retinopathy increased with increasing HbA1c levels. Approximately 17% of patients with diabetes whose HbA1c level was less than 7.0% had moderate-to-severe non-proliferative and proliferative DR. This number increased to approximately 34% in patients with HbA1c between 8.2% and 9.9%, and 50% in patients with a HbA1c level greater than 10%, demonstrating a high burden of diabetic retinal disease in this minority population [10].

#### 3.4.2. Hypertension

Elevated mean blood pressure, or hypertension, is another risk factor for DR [26]. A meta-analysis of 35 studies consisting of 22,896 individuals with diabetes found that patients with a blood pressure ≥140/90 mmHg had higher rates of DR than patients with blood pressure ≤140/90 mmHg (39.6% vs. 30.8%, respectively) [33]. In the Hoorn study, which consisted of 233 individuals with diabetes between the ages of 50 to 74 years, patients with hypertension had a more than double the risk of developing DR after 10 years when compared to patients without hypertension [34]. Additionally, several studies have demonstrated that intensive blood pressure therapy can decrease the progression of DR [31,35].

DR prevalence based on blood pressure has also been studied in minority populations. The LALES study found an increased odds ratio of 1.26 for developing DR for every 20 mm Hg increase in blood pressure [8,9]. However, in the Proyecto VER study, blood pressure, when adjusted for gender and age, was not significantly correlated with very early DR changes [10]. The MESA study found that elevated systolic blood pressure is a risk factor for DR, and that Black patients with diabetes were more likely to have hypertension [36]. The NHANES III study reported that there is a higher frequency of increased systolic blood pressure among Black patients with diabetes who develop DR. In another study of 380 African Americans with type 2 diabetes, Penman et al. also found increased systolic blood pressure to be a risk factor for the development of proliferative DR [37]. On the contrary, a study of 105 Black patients with type 2 diabetes found that known risk factors of DR, such as systolic blood pressure, did not explain the greater prevalence of DR in Black patients, and therefore suggested that genetic susceptibility may play a role in increased DR prevalence in Black populations [38]. Hypertension management may be targeted to prevent the development of vision-threatening disease in individuals with diabetes.

#### 3.4.3. Hyperlipidemia

Several studies in the United States have assessed the association between elevated lipid levels and DR. The DCCT found that patients with proliferative DR had higher total cholesterol, mean triglycerides, and low-density lipoprotein (LDL) levels [25]. Additionally, the total to HDL cholesterol ratio and LDL levels were found to predict clinically significant macular edema (CSME) in diabetic patients [25,34,39]. The Hoorn study also found serum cholesterol and triglycerides to be associated with DR prevalence [34]. On the other hand, the Cardiovascular Health Study (CHS) and MESA studies did not find an association between lipid levels and retinopathy [36,39]. Additionally, lipid-lowering therapies, such as fibrates and statins, in patients with diabetes and hyperlipidemia have shown mixed results as treatment for DR progression and macular edema [31,40,41].

#### 3.4.4. Nephropathy (Co-Morbidity)

Diabetic kidney disease (DKD) is the leading cause of chronic and/or end-stage kidney disease in the world, with a prevalence rate of approximately 40% in patients with diabetes [42]. Microalbuminuria and an elevated albumin-to-creatinine ratios are indicators of nephropathy, which are also associated with a greater risk of developing DR [43,44]. The Atherosclerosis Risk in Communities (ARIC) also found that renal dysfunction, as measured by microalbuminuria and elevated albumin-to-creatinine ratios, was associated with retinopathy, and this association persisted independent of age, presence of diabetes, hypertension, and other risk factors [44]. The Trial to Reduce Cardiovascular Events with Aranesp Therapy (TREAT) reported that 47% of 4038 participating patients, who all had type 2 diabetes, renal dysfunction, and anemia, also had DR [45]. In patients with diabetes, the link between retinopathy and nephropathy may be explained by chronic hyperglycemia, which can result in microvascular damage in both the retina of the eye and the glomerulus of the kidney. Racial and ethnic minority patients with diabetes are reported to have a higher prevalence of nephropathy, including end-stage renal disease [42,46]. The New Jersey 725, consisting of all African American participants, reported that 49.8% of patients with DR also had renal disease [47].

### 3.5. Screening for Diabetic Retinal Disease

#### 3.5.1. Screening Recommendations

To prevent vision loss and blindness from diabetic retinal disease, guidelines emphasize routine screening and early detection of diabetic eye disease [48]. The American Diabetes Association (ADA) and the American Academy of Ophthalmology (AAO) recommend that adults with type 1 diabetes should have an initial dilated eye exam within 5 years of diabetes diagnosis [3]. If there is no evidence of diabetic retinopathy after one or more annual eye exams, then diabetic retinopathy screening can be considered every 2 years [3]. Diabetic retinopathy screening is recommended at the time of diagnosis for type 2 diabetic patients and annually after diagnosis [49]. Only 11–71% of individuals with diabetes complete the recommended diabetic eye screenings [50,51,52,53]. While there are many known barriers to diabetic retinopathy screening [7,36,54,55], disparities due to social determinants of health further exacerbate suboptimal screening rates.

#### 3.5.2. Socioeconomic Status (Income and Education)

Lower socioeconomic status and ethnic minority status are associated with decreased health care use, decreased access to eye care, lower adherence to diabetic eye exams, and worse health outcomes [56,57,58]. A large U.S study of Medical Expenditure Panel Survey (MEPS) data from 2002–2009 found that individuals with private or public insurance coverage were more likely to have diabetic eye exams than uninsured individuals [59,60]. Other studies have shown that despite availability of screening, there is under-utilization of diabetic eye exams among safety-net minority communities, particularly African American and Hispanic populations, and those disadvantaged in employment status, household income, and education [61].

Lack of education around the need for annual diabetic eye exams, miscommunication, or lack of communication with providers, and no perceived need for diabetic retinopathy screening are well described educational and health literacy barriers to diabetic retinopathy screening [57,62]. In a qualitative survey study of participants with type 2 diabetes, the researchers found that patients were more likely to seek eye care if they had visual problems or concerns and may not have had their regular diabetic eye exams if not for the visual concerns [62]. Participants in this study reported a range of knowledge about DR, with some participants understanding that blindness can result from DR and others only expressing that they knew vision could be affected in some way [62]. The results of interviews indicated a gap in patients’ understanding of DR and the utility of preventative eye care [62]. Another cited barrier is miscommunication between the patient and provider, with a lack of clear communication regarding the recommendation for a diabetic eye exam, as well as confusion between routine eye care and diabetic eye examination [63,64,65]. Furthermore, in a focus group study performed by Elam et al., patients reported wanting clearer communication of their diabetes care plan to better understand management [66]. Limited health literacy is also associated with suboptimal diabetes control and lower utilization rates of preventive services [57]. A study utilizing the health literacy assessment in adults with type 2 diabetes followed in two primary care settings demonstrated that inadequate health literacy was associated with suboptimal glycemic control, as well as an increased risk for diabetic retinopathy [64]. Interactive physician-patient communication was associated with improved glycemic control, suggesting that implementation of consistent interactive education overcomes communication barriers and leads to better glycemic outcome [67].

In the UK, which was the first country to offer systematically organized DR screening for all patients with diabetes over the age of 12 years, reasons for screening non-compliance were evaluated and found to include patients forgetting to get screened, being anxious about getting screened, being misinformed about the purpose of screening, and general disengagement with their diabetes care [68]. Studies have also shown that patient education can directly improve screening; in a cohort of 101 patients with diabetes in Southern Ontario, baseline screening adherence increased from 35 to 50% upon implementation of an educational DR seminar, in addition to fundus photography at screening locations [69,70].

#### 3.5.3. Geography (Neighborhood and Physical Environment)

Access to DR screening based on geographic location, accessibility of eye care centers, and transportation also impacts screening adherence rates. Rural populations are less likely to complete DR screening due to long travel distances for care, costs associated with time off from work and travel, and infrequent healthcare access [53]. Even in urban settings, where public transportation is available, patients cited transportation as a barrier to diabetic eye care [63].

Differences in screening can additionally be attributed to one’s housing environment. Racial segregation is a key cause to the disparities seen in healthcare due to the link to socioeconomic status segregation [71,72,73]. Those living in neighborhoods with lower socioeconomic status often experience disparities in access to recommended health services [74]. A recent study performed in the Washington DC region assessed social determinants of health and adherence to recommended diabetic eye exams. In this cohort, almost 50% of participants completed their recommended diabetic eye exams, but in individuals who did not, lack of a primary care provider and poor housing conditions were both associated with lower adherence to recommended DR screening [75].

#### 3.5.4. Healthcare Access and Affordability

In a study of adults with T1D, low socioeconomic level, based on neighborhood-level measures of socioeconomic deprivation, was associated with the development of DR [17]. Patients were less successful in trying to obtain eye care appointments with Medicaid than with private insurance, suggesting a disparity in access to eye care based on insurance status [56]. Improving access to eye care professionals for those with Medicaid may improve health outcomes and decrease health care spending in the long term [56,75].

In a study evaluating the availability of care appointments for patients with Medicaid in rural areas of Maryland and Michigan, the authors found that due to the small number of eye care professionals in rural areas, patients with Medicaid had to travel long distances for eye care [56]. In general, Medicaid patients have lower rates of completing annual diabetic eye exams [76], and this likely exacerbates already low screening rates [77]. Further, even in federally funded community health care centers, only 20% have an onsite eye care professional, further exacerbating the lack of access to eye care [78].

Adherence to regular DR examinations has been reported to be as low as 11% in adults [79], but implementation of universal screening programs, teleretinal screening programs, and more recently, AI screening programs, have shown success in improving screening rates and thereby reducing the risk of blindness from DR [80,81,82,83].

In the US, a study implementing teleretinal screening in rural and underserved areas of North Carolina demonstrated an increased eye screening rate from 9.3% to 60.0% [83,84]. Additionally, teleophthalmology screening identified higher rates of diabetic eye disease in individuals seen in county Safety Net Hospitals (SNHs) compared with non-SNHs. These results demonstrate the utility of teleretinal screening for diabetic eye disease in resource limited settings in optimizing screening and prevention of DR where disparities in prevalence and screening adherence exist [85]. In a study assessing participants’ satisfaction with telemedicine eye screening, 90.8% of patients found telemedicine a satisfactory alternative to an eye care provider exam [82]. Additionally, 82% of patients considered telemedicine faster, easier, more accessible, and convenient as their reasoning for their preference [82]. Due to the ability of teleophthalmology to mitigate health inequities and address barriers faced by underserved communities, it remains a vital tool in DR screening.

In 2018, the FDA approved the first autonomous artificial intelligence system for diabetic retinopathy screening, where AI-based algorithms are used to detect DR from retinal images taken by a non-mydriatic fundus camera, providing an immediate result at the point of care. These systems have high diagnostic accuracy both in clinical trials and in real-world settings, demonstrating high sensitivity and specificity of 87.2% and 90.7%, respectively [3,81,86,87]. After the implementation of autonomous AI, DR screening adherence improved from 49% to 95% in a diverse pediatric population [70]. Additionally, patient satisfaction with automated screening is high, with 96% of patients reported as being satisfied or very satisfied with this method [88]. Autonomous artificial intelligence DR screening has also been demonstrated to be cost-saving for the patient, family, and health care system [89,90,91] Diagnostic AI for retinopathy screening has the potential to improve access to screening, mitigate disparities, and reduce health care costs for patients and health care systems on a large scale.

### 3.6. Management and Outcomes for Diabetic Retinal Diseases

In addition to disparities in prevalence and screening in relation to social determinants of health, there also exist disparities in the management and outcome of DRD. Focal laser photocoagulation has historically been used as a frontline treatment for eyes with clinically significant macular edema, after the Early Treatment Diabetic Retinopathy Study (ETDRS) showed that treated eyes had a substantially reduced risk of vision loss [86,92]. However, in recent years, the Protocol I study by the Diabetic Retinopathy Clinical Research Network (DRCR) established the higher efficacy of anti-VEGF therapy over focal laser treatment, cementing anti-VEGF injections as frontline therapies in the management of DME [93]. The 2-year follow-up results of the Protocol T study suggested that in patients presenting with lower baseline visual acuity, bevacizumab therapy resulted in smaller gains in visual acuity than aflibercept or ranibizumab [93]. Other recent studies reported that Hispanic, Black, and Medicaid-insured patients have worse DR severity and visual acuity (VA) at initiation of anti-VEGF treatment for DME [94], and that treatment-naive Black patients receiving intravitreal bevacizumab experience lower odds of VA improvement after the first and third injection [95], suggesting both a racial and socioeconomic disparity in the pharmacological treatment of DME. While racial and ethnic disparities have not been seen in treatment of commercially insured patients with diabetic retinopathy [96], other factors, such as having a co-pay, have reportedly lowered the odds of a patient receiving any treatment for DME (although having a high deductible and type of insurance plan were not associated with initiating treatment) [97].

There are many reports suggesting that minority and low-income individuals are underrepresented in clinical trials of treatments for diabetic retinopathy. Black patients, especially, are vastly under-represented in DME clinical trials, with one study reporting that Black patients with DME are 3 times less likely to be represented in NIH trials and 4.5 times less likely to be represented in industry trials for DME compared to their representation in the US population with DME [98,99,100]. In clinical trials, adherence to therapies may be supported by intensive efforts by the study team as well as financial compensation to the participant, which are not feasible in regular clinical practice. Trials designed to ascertain whether the treatment may “work” often achieve internal validity at the expense of uncertainty about generalizability, since the populations enrolled may differ drastically from those seen in clinical practice [101]. Furthermore, interventional clinical trials for diabetic retinopathy are often conducted in large, urban, tertiary, or quaternary referral centers, and patients residing in areas that are more rural, more impoverished, and from regions outside the Northeast in the United States were likely under-represented in these clinical trials [102].

Patients receiving real-world treatment for DME tend to follow-up less than in clinical trials and show fewer improvements in visual acuity than patients in clinical trials using the same anti-VEGF agents; real-world studies and a sub-analysis of protocol T showed an association between missed follow-up and worse visual outcomes [103,104,105]. In patients undergoing anti-VEGF treatment for DME, several factors are associated with loss to follow-up, including Hispanic ethnicity, residing in communities with lower average income, having government-provided insurance, and having lower visual acuity [94,106,107]. Efforts to recruit study participants who are typically underrepresented can help determine the impact of new DR treatments in all people with diabetes, while promoting health and equity.

## 4. Conclusions

As the prevalence of diabetes in the U.S. and worldwide continues to increase, as does the burden of diabetes-related complications, particularly diabetic retinopathy and macular edema, it is important to address inequities due to SDOH in the prevalence, screening, management, and outcomes of DRD. Lower socioeconomic status, household income, education, rural geography, and minority status are all associated with reduced access to diabetic retinopathy screening. Delays in diagnosis compounded with risk factors for diabetic retinopathy portends worse visual outcomes. Implementation of accessible DR screening programs using teleretinal networks or autonomous artificial intelligence-based screening in underserved and under-resourced rural and low-income areas can improve early detection of DR, thereby reducing poor visual outcomes in these communities. The diagnostic accuracy achievable by AI paired with its scalability has the potential to increase screening access and diagnosis, and mitigate disparities related to social determinants of health, provided that diagnosability, racial and ethnic bias, explainability, and data usage are adequate addressed in the development. Early detection and diagnosis of DR can also improve long-term visual outcomes for the millions of individuals with diabetes in the US. Although there have been significant advancements in the management of DRD, offering equitable therapies and optimizing adherence where disparities exist can help improve visual outcomes. Future research should aim to increase generalizability in clinical trials with a greater representation of minority populations in study cohorts, as well as the evaluation of visual outcomes further down the clinical pathway. Efforts to promote equity in diabetes clinical care should focus on mitigating care gaps related to social determinants of health to ensure optimal visual outcomes for all people with diabetes.

There are limitations to this narrative review in that publications reviewed focused on the US health care system; therefore, they do not address global concerns around inequities and social determinants of health associated with diabetic retinopathy.

## Data Availability

All data is available in the published literature.

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
