# Peer review of "Social Determinants of Health and Impact on Screening, Prevalence, and Management of Diabetic Retinopathy in Adults: A Narrative Review"

_jcm, 2022, doi:10.3390/jcm11237120_

Round 1

Reviewer 1 Report

This manuscript is a narrative review on DR vs. social determinants of health. The authors report recent literature pertaining to this topic, which highlights the reported associations of various SDoH on DR screening and its clinical management.

The review is a narrative one. This topic which is not new in itself, could have benefited from a more rigourous methodolody (scoping or systematic review) in order to highlight potential novelties, which unfortunately this manuscript falls short of doing.

 It is also unfortunate that the manuscript lacks some structure elements which could help the reader better situate certain key elements and make the reading more inviting. Namely, besides the Introduction and Conclusion, there are no large subdivisions in the text.  A methods section should be included, along with search algorithm employed.  Although this is a narrative review (and not a scoping nor systematic review), the readers should be informed where the data presented are coming from and how were they sought. In addition, the manuscript is a rather lenghty enumeration of study results, which of course is appropriate for a narrative review, but lacks a true discussion section, where authors may highlight and discuss key findings (especially recent one, as this topic is not new) and their potential impacts or policy implications.  This is briefly touched upon in Conclusion section, but should be much more expanded, if the authors wish to bring to light something novel.

Other comments:

- change formulation of "race" for "ethnicity" (L49, L76, L152, L250 and others).

- L86: highlight that this is in the UK, despite universal health coverage 

- L106: remove underline style

Author Response

To the editorial team:

Thank you for taking the time to review our manuscript jcm-1996747 entitled: Social determinants of health and impact on screening, prevalence and management of diabetic retinopathy in adults: A narrative review.”

We have made changes to our manuscript as per the reviewer’s helpful recommendations. Our responses to the reviewers’ comments are bolded below.

Sincerely,

Risa Wolf & Co-authors

REVIEWER COMMENTS:

Reviewer 1:

This manuscript is a narrative review on DR vs. social determinants of health. The authors report recent literature pertaining to this topic, which highlights the reported associations of various SDoH on DR screening and its clinical management. The review is a narrative one. This topic which is not new in itself, could have benefited from a more rigourous methodolody (scoping or systematic review) in order to highlight potential novelties, which unfortunately this manuscript falls short of doing.

Response: Thank you for this helpful comment, we have added a methods section with details on the search terms employed.

A literature search was conducted through Pubmed, MEDLINE, EMBASE, and Scopus from February 2022 to September 2022. The following keywords and terms were used to identify articles for review and inclusion: screening, diabetic retinopathy, diabetic retinal disease, diabetic macular edema, prevalence, management, income, education, clinical trials, disparities, social determinants of health, and access. Since the focus of the narrative review was on social determinants of health in the U.S., all included articles were in English.”

 It is also unfortunate that the manuscript lacks some structure elements which could help the reader better situate certain key elements and make the reading more inviting. Namely, besides the Introduction and Conclusion, there are no large subdivisions in the text.  A methods section should be included, along with search algorithm employed.  Although this is a narrative review (and not a scoping nor systematic review), the readers should be informed where the data presented are coming from and how were they sought. 

Response: We have added more subdivisions in the text to make it easier to read. We have also added a methods section as described above.

In addition, the manuscript is a rather lenghty enumeration of study results, which of course is appropriate for a narrative review, but lacks a true discussion section, where authors may highlight and discuss key findings (especially recent one, as this topic is not new) and their potential impacts or policy implications.  This is briefly touched upon in Conclusion section, but should be much more expanded, if the authors wish to bring to light something novel.

Response: We have expanded the conclusion section to include a more thorough discussion.

Other comments:

  • change formulation of "race" for "ethnicity" (L49, L76, L152, L250 and others).

Response: Complete

  • L86: highlight that this is in the UK, despite universal health coverage 

Response: Complete

  • L106: remove underline style

Response: Complete

Reviewer 2 Report

To the authors to present the “Social determinants of health and impact on screening, prevalence 2 and management of diabetic retinopathy in adults: A narrative review”

We are facing a work that, although it provides information on the detection and prevention of Diabetes and its implications at the ocular level, I do not find it well-structured or that the information it provides is scientifically relevant, so I dare to suggest some changes:

#1 In the structure of the paper there is an epigraph, introduction, and it seems that everything is part of it until reaching the conclusions.

There is no clear structure. To rewrite.

# 2 There are very old bibliographic citations, some from more than 20 years ago: 93, 72, 68, 65, 56, 49, 48, 40, 39, 35, 33, 31, 26, 11, 10.

# 3 I would find more interesting a systematic review and confront data from other countries with different health care systems.

Author Response

To the editorial team:

Thank you for taking the time to review our manuscript jcm-1996747 entitled: Social determinants of health and impact on screening, prevalence and management of diabetic retinopathy in adults: A narrative review.”

We have made changes to our manuscript as per the reviewer’s helpful recommendations. Our responses to the reviewers’ comments are bolded below.

Sincerely,

Risa Wolf & Co-authors

REVIEWER COMMENTS:

Reviewer 2:

To the authors to present the “Social determinants of health and impact on screening, prevalence 2 and management of diabetic retinopathy in adults: A narrative review”

We are facing a work that, although it provides information on the detection and prevention of Diabetes and its implications at the ocular level, I do not find it well-structured or that the information it provides is scientifically relevant, so I dare to suggest some changes:

#1 In the structure of the paper there is an epigraph, introduction, and it seems that everything is part of it until reaching the conclusions.

There is no clear structure. To rewrite.

Response: Thank you for this helpful feedback. We have addressed this as detailed above for Reviewer #1.

# 2 There are very old bibliographic citations, some from more than 20 years ago: 93, 72, 68, 65, 56, 49, 48, 40, 39, 35, 33, 31, 26, 11, 10.

Response: Thank you for highlighting this. Some of these references were felt to be sentinel articles and thus included. We have eliminated the following to make the review more up to date: 31, 40, 49, and 72

# 3 I would find more interesting a systematic review and confront data from other countries with different health care systems.

Response: This is an excellent suggestion. While this is beyond the scope of this current narrative review, we agree this would be a very interesting comparison and systematic review.

Round 2

Reviewer 1 Report

This second version of this narrative review on DR vs. SDoH adressed all of the comments in the first round of review.

In the conclusion section, it would be relevant to add a quick recap(1-2 sentences) of the major findings of the review (i.e. the major associations of SDoH vs. DR), around L419 (after "...DRD" and before "Implementation...").

Also, on L423-424 (as well as earlier, in L365-66), the comments on AI are quite absolute (diagnostic accuracy of AI, infinite scalability, etc.) and should be more nuanced, especially since this is not supported by the rest of the study results.  The limitations to AI screening programs no doubt also exist in the literature and should be included (or at least, the tone of the sentence should be more conservative with regards to the potential of AI - AI can serve as a complementary tool to current systems, etc.).  This is probably especially true for underserved and remote communities (Indigenous, etc.), where image acquisition is not always optimal if technical staff has a lot of turnover.

Finally, 1-2 sentences on study limitations should be added towards the end of the discussion.  

Author Response

Thank you for taking the time to re-review our manuscript jcm-1996747 entitled: Social determinants of health and impact on screening, prevalence and management of diabetic retinopathy in adults: A narrative review.”

We have made changes to our manuscript as per the reviewer’s helpful recommendations. Our responses to the reviewers’ comments are bolded below.

Sincerely,

Risa Wolf & Co-authors

This second version of this narrative review on DR vs. SDoH adressed all of the comments in the first round of review.

Response: Thank you

In the conclusion section, it would be relevant to add a quick recap(1-2 sentences) of the major findings of the review (i.e. the major associations of SDoH vs. DR), around L419 (after "...DRD" and before "Implementation...").

Response: Thank you for this suggestion to enhance the conclusion section, we have added the following sentence:

“Lower socioeconomic status, household income, education, rural geography and minority status are all associated with reduced access to diabetic retinopathy screening. Delays in diagnosis compounded with risk factors for diabetic retinopathy portends worse visual outcomes. “

Also, on L423-424 (as well as earlier, in L365-66), the comments on AI are quite absolute (diagnostic accuracy of AI, infinite scalability, etc.) and should be more nuanced, especially since this is not supported by the rest of the study results.  The limitations to AI screening programs no doubt also exist in the literature and should be included (or at least, the tone of the sentence should be more conservative with regards to the potential of AI - AI can serve as a complementary tool to current systems, etc.).  This is probably especially true for underserved and remote communities (Indigenous, etc.), where image acquisition is not always optimal if technical staff has a lot of turnover.

Response: We removed the statement about infinite scalability of AI in both sections as it is not supported by the literature. Most published reports on AI, however, are related to diagnostic accuracy so we left that text intact.

Finally, 1-2 sentences on study limitations should be added towards the end of the discussion.  

Response: We have added a limitations statement, thank you for this suggestion.

“ There are limitations to this narrative review in that publications reviewed focused on the US health care system, and do not address global concerns around inequities and social determinants of health associated with diabetic retinopathy.“

Reviewer 2 Report

Thank you for reviewing your article.

I continue to suggest that it would add great value to the work to carry out a systematic review.

Author Response

Thank you for taking the time to re-review our manuscript jcm-1996747 entitled: Social determinants of health and impact on screening, prevalence and management of diabetic retinopathy in adults: A narrative review.”

Reviewer comment:

I continue to suggest that it would add great value to the work to carry out a systematic review.

Response: Thank you for the suggestion. We agree that a systematic review would add to the existing literature, but is beyond the scope of this current narrative review.